# Graph network simulators can learn discontinuous, rigid contact dynamics

**Kelsey R. Allen**[*], **Tatiana Lopez-Guevara**[*],
**Yulia Rubanova, Kimberly Stachenfeld, Alvaro Sanchez-Gonzalez,**
**Peter Battaglia, Tobias Pfaff**
DeepMind, UK

**Abstract:** Recent years have seen a rise in techniques for modeling discontinuous dynamics, such as rigid contact or switching motion modes, using deep learning. A common claim is that deep networks are incapable of accurately modeling rigid-body dynamics without explicit modules for handling contacts, due to the continuous nature of how deep networks are parameterized. Here we investigate this claim with experiments on established real and simulated datasets and show that general-purpose graph network simulators, with no contact-specific assumptions, can learn and predict contact discontinuities. Furthermore, contact dynamics learned by graph network simulators capture real-world cube tossing trajectories more accurately than highly engineered robotics simulators, even when provided with only 8 – 16 trajectories. Overall, this suggests that rigid-body dynamics do not pose a fundamental challenge for deep networks with the appropriate general architecture and parameterization. Instead, our work opens new directions for considering when deep learning-based models might be preferable to traditional simulation environments for accurately modeling real-world contact dynamics.

**Keywords:** graph networks, contacts, rigid body dynamics

## 1 Introduction

Model-based control relies on the ability to predict how objects in the world are going to be affected by an agent's actions. Considerable effort has therefore been invested into building precise analytical models of physical dynamics, such that these can be used effectively and efficiently for safe planning and control. However, some effects such as frictional contacts or temperature-dependence in actuators are hard to model or to estimate via system identification, resulting in significant drift when such analytical models are used to predict real-world dynamics (the "sim-to-real gap"). To counteract these problems, the last few years have seen increased interest in *learned* dynamics models which can be directly fit to real data [1, 2], run on hardware accelerators to significantly speed up policy, model training and inference [3, 4], and may side-step issues of inferring state representations by operating directly at the pixel level [5, 6, 7].

With the increased attention on learned models, there is an incentive to understand which kinds of models can accurately capture the different types of dynamics a robot may encounter. In particular, many tasks in robotics, from locomotion to manipulation, require making and breaking contact with rigid objects. This phenomenon is known to be discontinuous [8, 9], as the spatial and temporal scales on which rigid bodies deform during contact are imperceptibly small. Recent work has suggested that modelling contacts presents a fundamental challenge for deep learning models because of their tendency toward smooth interpolation [10, 11], but also even for analytic robotics simulators when compared to real-world data [12, 1].

To remedy these issues with rigid-body dynamics learning, several papers have proposed combining deep learning with analytic rigid-body dynamics models to varying degrees [1, 2, 11, 13, 14, 15]. While such techniques can improve stability and prediction accuracy in some cases, they also come

---

[*]These authors contributed equally. Correspondence to {`krallen,zepolitat`}`@deepmind.com`

6th Conference on Robot Learning (CoRL 2022), Auckland, New Zealand.

with some of the same drawbacks as analytical solvers: the analytical components may pose restrictions on the type of objects and dynamics that can be simulated, may be hard to parameterize correctly, or may induce strict time-stepping limits. This precludes the more general use of these models in a wider variety of physical phenomena, such as interactions between rigid bodies and fluids or deformable objects, and might hamper generalization to more complex, real-world situations in which e.g. contact forces are not perfectly preserved.

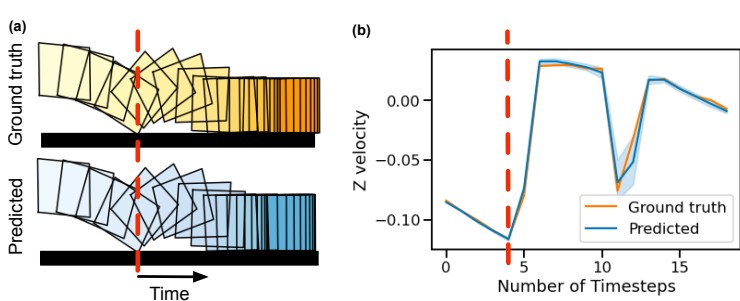

Figure 1: (a) A 2D projection of a sample 3D trajectory from Pfrommer et al. [2] of a cube being tossed onto a table (orange) and the predictions obtained by a Graph Network Simulator (GNS) (blue). The cube contacts the table along an edge, bouncing into the air. The red line shows the time of contact. (b) The GNS is capable of representing and correctly resolving the velocity discontinuity introduced by the rigid contact event (dashed red line), closely matching the ground truth. The shaded area shows 95% confidence interval across 10 seeds.

Here we show that there are simple, general ways to use deep learning to accurately model discontinuous contact dynamics without specific rigid contact parameterizations, and which lead to better performance for real-world trajectory modeling. We find that graph network simulators [16, 3], are sufficient to capture rigid body dynamics on both simulated and real world data with excellent sample efficiency, outperforming learned models with explicit handling of rigid contacts [2]. For modeling real world cube tosses, graph network simulators can even outperform carefully tuned analytic simulators common in robotics, such as Drake [17], Bullet [18], and MuJoCo [19]. We perform carefully controlled experiments on a variety of rigid-body dynamics environments, and demonstrate the specific architectural choices for graph network simulators that support discontinuous dynamics learning. These results demonstrate that explicit modeling of rigid contacts is not necessary for deep learning methods to handle discontinuous dynamics; instead, we find that general-purpose, relational, translation-equivariant models are equally or more capable of handling these events precisely.

## 2 Related work

Dynamics with discontinuities, such as rigid-body collisions or frictional transitions, are challenging to learn due to the non-smooth and discontinuous nature of the dynamics involved [2, 10, 1]. Classical approaches model such behaviour via rigid [8] or approximately rigid contact models [20, 21, 22]. However, there is clear evidence that these methods struggle to capture the behaviour of real world rigid impacts [1, 12, 23].

Another approach is to learn a function that approximates the rigid-body dynamics directly from data. For example, Gaussian processes (GPs) are a popular technique that has been used to control robotic arms [24], learn slip-stick transitions [25] and model planar contact [1]. GPs are data-efficient but include smoothness assumptions that do not generalize well to discontinuous dynamics.

To better capture discontinuities while still leveraging access to data, a variety of recent methods use machine learning to augment analytic physics simulators. Some techniques parameterize contact dynamics as a linear complementarity program, and provide differentiable solvers that can be coupled with deep networks [13, 26]. Others incorporate a non-differentiable physics engine to make predictions, but also learn a residual model to augment those such that they more closely reflect observed dynamics [1, 14, 15]. Another class of approaches uses implicit models that embed analytic equations and solvers within end-to-end techniques [2, 11]. In all cases, these techniques explicitly encode rigid-body interactions, which limits their application to richer physical phenomena.

Over the past several years, Graph Neural Networks (GNNs) have been increasingly applied to physical simulation. These models explicitly learn interactions between entities parameterized with

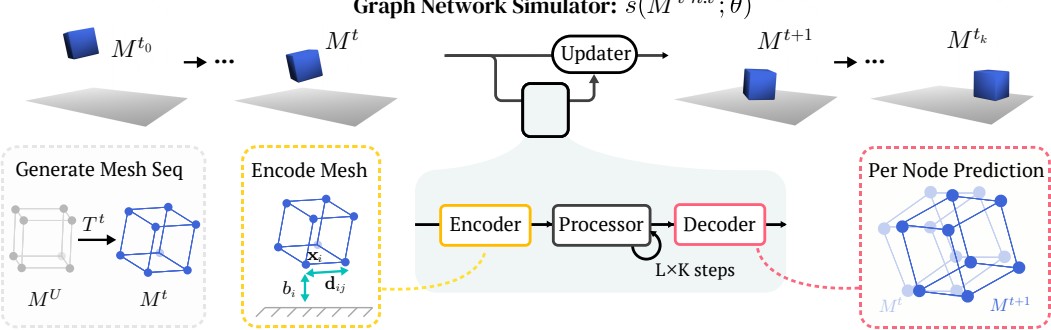

Figure 2: Graph Network Simulator (GNS) model for rigid shapes. We construct the input meshes $M^{t_k}$ by applying the object center of mass's translation and rotation $T^{t_k}$ to the undeformed object shape $M^U$. The encoder build a graph from $M^{t_k}$, and encodes relative positional features into graph nodes and edges. Next, the processor computes $L \times K$ message passing steps on this graph. The decoder outputs accelerations $a_i$ for each node, which are integrated into updated node positions.

neural networks. Early applications of GNNs modeled interactions between simple rigid entities like bouncing billiards [27], and later work extended this to more complex domains like pushed and falling shapes [28, 29], moving robot bodies [30, 31, 32], and even fluids and deformable meshes [33, 16, 3]. Despite the successes of these models in many areas, there have been no careful analyses of how they perform in settings with hard discontinuities. Furthermore, most results are shown only in simulation relative to some ground truth simulator, which are themselves often inaccurate [1, 12]. Exacerbating this concern, recent papers have implied that there are simple cases where deep networks fail to learn accurate collision dynamics [34, 10, 11]. These results cast doubt on whether end-to-end deep learning models, including GNNs, are generally a good fit for learning rigid-body interactions, particularly if the rigid bodies are complex, volumetric objects like cubes or cylinders.

In this work, we show for the first time that GNN-based simulators, constructed with simple design choices, are capable of capturing real-world rigid contacts for some object shapes as effectively or better than analytic models [19, 18, 17] and learned models with explicit contact handling [2]. We conduct a series of ablations and experiments to shed light on exactly which properties of GNNs enable this high performance.

## 3   Approach

A rigid-body simulation, or real-world experiment, can be described by the object shapes (e.g. represented by an undeformed mesh $M^U$ with nodes corresponding to object vertices, and edges corresponding to object faces), and sequence of rigid transformations $\mathcal{T} = (T^{t_0}, \dots, T^{t_k})$ consisting of the translation and rotation at each time step $t$. Here we consider a GNN-based approach to simulation, trained using sequences of meshes $\mathcal{M} = (M^{t_0}, \dots, M^{t_k})$, where each mesh consists of the absolute position $\{\mathbf{x}_i\}$ for each node, as well as edges connecting these nodes. We can obtain such a sequence by applying $\mathcal{T}$ to the initial object shape $M^U$ as $(T^{t_0}(M^U), \dots, T^{t_k}(M^U))$.

### 3.1   Learning Forward Dynamics

We define the Graph Network simulator (GNS) model $s_\theta$, which given $M^t$ and a short history of states $\{M^{t-1}, \dots, M^{t-h}\}$ (we use $h = 2$), outputs a next state mesh prediction $\tilde{M}^{t+1}$. This model can be iteratively applied for $k$ steps to predict a trajectory $(M^{t_0}, \tilde{M}^{t_1}, \dots, \tilde{M}^{t_k})$ (Figure 2). $s_\theta$ is represented by an Encode-Process-Decode GNN [3, 16, 35], adapted to rigid bodies. Below we describe the model components. We specifically consider the case of an object interacting with the floor, but multi-object collisions can be treated analogously using world edges as in [3].

**Encoder**   The encoder constructs a graph $\mathcal{G} = (\mathcal{V}, \mathcal{E})$ from the input mesh $M^t$, using the mesh vertices as nodes $\mathcal{V}$, and mesh edges as graph edges $\mathcal{E}$. Each node has features $V_i = \{\mathbf{v}_i^{\text{FD}}, b_i\}$, where $\mathbf{v}_i^{\text{FD}} = (\mathbf{x}_i^t - \mathbf{x}_i^{t-1}, \dots, \mathbf{x}_i^{t-h+1} - \mathbf{x}_i^{t-h})$ is the node's finite-difference (FD) velocity over the input history, and $b_i$ is the distance to the domain boundaries (here, the floor). The boundary distance $b_i$ is

clipped to the range $[0, 0.5]$ as in [16], to prevent overfitting for states far away from contact. An edge from node $i$ to $j$ has features $e_{ij} = \{\mathbf{d}_{ij}, ||\mathbf{d}_{ij}||, \mathbf{d^U}_{ij}, ||\mathbf{d^U}_{ij}||\}$, where $\mathbf{d}_{ij} = \mathbf{x}_i - \mathbf{x}_j$ is a vector of the relative displacement between the nodes $i$ and $j$. Similarly, $\mathbf{d}_{ij}^U$ denotes the displacement between these nodes on the undeformed initial mesh $M^U$, which can help the model to retain the rigid shape. We note that this is a local and translation-equivariant encoding of the simulation state, which is particularly important in the data-sparse setting. We also consider the commonly used absolute position encoding [33, 29, 28, 30], in which $V_i = \{\mathbf{v}_i^{\text{FD}}, \mathbf{x}_i, \mathbf{x}_i^U\}, E_{ij} = \{\}$, which does not have these properties. All node and edges features, respectively, are concatenated and passed to encoder multi-layer perceptrons (MLPs) that produce latent vectors of size 128 for each node and edge in the graph.

**Processor** After encoding features, the processor computes several steps of message-passing amongst the nodes in the graph as in [16]. In each message-passing step, each latent edge $E_{ij}$ and node feature $V_i$ is updated to $E'_{ij}$ and $V'_i$ respectively according to:

$$E'_{ij} \leftarrow \rho_\ell^E(E_{ij}, V_i, V_j), \quad V'_i \leftarrow \rho_\ell^V(V_i, \sum_j E'_{ij}) \tag{1}$$

with $\rho_\ell^E$ and $\rho_\ell^V$ implemented as MLPs. A message passing block consists of $L$ message passing updates with unshared weights for all $\rho_\ell$. Each step, this block is repeated $K$ times, resulting in $L \times K$ updates with $\mathcal{O}(L)$ weights.

**Decoder** The decoder MLP $\delta^V$ decodes the node latent features after the last message passing step to predict the per-node finite-difference acceleration estimate $\mathbf{a}_i$ for each node $i$. Compared to predicting absolution position, this is a soft bias for inertial dynamics.

**Model training and data augmentation** We compute the loss as the *per-node* mean-squared-error on acceleration prediction, and train the model using the Adam optimizer. The ground-truth acceleration is estimated from the position data using finite differences. While the model is translation-equivariant by design, we can exploit an additional symmetry of the problem: during data loading of a training pair, we rotate both input and target around the $z$ (up) direction with an angle uniformly sampled from $[-\pi, \pi]$. This decreases overfitting without requiring additional data.

**Updater** To generate simulation trajectories at test time, the model is applied iteratively. After each step, the updater integrates the per-node acceleration predictions with an Euler integrator to compute the next-step position as $\mathbf{x}_i^{t+1} = \mathbf{a}_i + 2\mathbf{x}_i^t - \mathbf{x}_i^{t-1}$. For long rollouts, positional error can accumulate and deform the object shape. As the only architectural distinction from Pfaff et al. [3], we prevent this by using shape matching [36] during rollout. That is, after each step we fit a transformation to the predicted node positions, and re-project the nodes to the transformed mesh to enforce shape consistency (see Appendix B for details). We do not use shape matching during training and compute the loss directly on network predictions.

## 4 Results

Rigid-body dynamics introduce two unique challenges for learning dynamics models: (1) discontinuous dynamics during rigid collisions, and (2) extreme sensitivity to initial conditions. Our experiments in this section provide a qualitative evaluation of the capabilities of GNS models to handle the aforementioned problems, as well as quantitative comparisons with models encoding rigid contact-specific priors. For videos of rollouts and access to code, please see https://sites.google.com/view/gnn-rigids/home.

### 4.1 Diagnostic qualitative results

We first perform a qualitative evaluation of whether a GNS can capture discontinuous dynamics and its sensitivity to initial conditions. The GNS in Figure 1 was trained on 256 real cube tossing trajectories from [2], while the GNS for Figure 3 was trained on 8096 simulated MuJoCo trajectories from [10]. All figures show projections of 3D trajectories. See details on the training data in Appendix A.

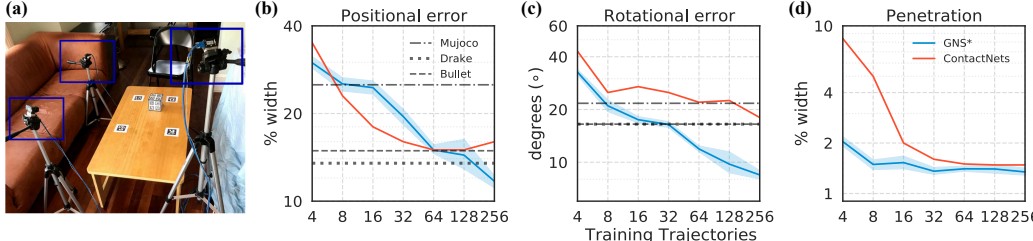

Figure 4: (a) Real world cube throwing dataset (Figure 3 from Pfrommer et al. [2]). (b) - (d): performance metrics from [2] indicating positional error, rotation error, and predicted penetration between the cube and the table. Lower is better. GNS (blue) is more accurate than the ContactNets-Polytope method [2] (red) for $> 64$ training trajectories, and outperforms simulation with analytical solvers with between 8 and 128 trajectories (dotted lines).

**Discontinuous dynamics**   Consider the scenario shown in Figure 1(a). In this simple setup, a cube is thrown onto the ground such that it contacts the floor along one of its edges. This contact produces a sharp discontinuity in the velocity as the cube bounces upwards. In Figure 1(b), the graph network simulator is able to capture the discontinuities on each bounce accurately, matching the real data closely (even across 10 separate training seeds). This runs counter to prevailing belief (e.g. [10]) which suggests that deep networks struggle to learn such discontinuities.

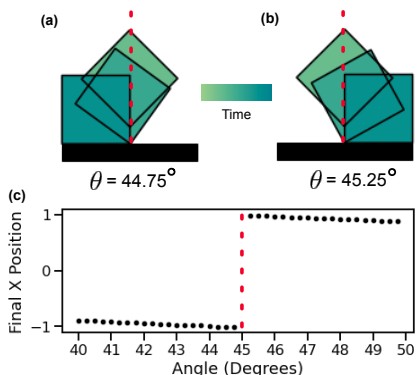

Figure 3: Graph Network Simulators learn to be appropriately sensitive to initial conditions, including capturing discontinuities. A cube is correctly predicted to fall to the left when angled $< 45°$ respective to the floor, and right for angles $> 45°$.

**Sensitivity to initial conditions**   Next, consider the scenario shown in Figure 3. In this setup, a cube is falling directly downwards with a rotation $\theta$ close to $45°$. For any angle $\theta < 45°$, the cube will fall to the left, while for any angle $\theta > 45°$, the cube will fall to the right. Here the model must be acutely sensitive to rotations differing by fractions of a degree, as these will significantly affect the final position of the cube. We see that the graph network simulator closely captures the discontinuous behavior in this diagnostic scenario: even for barely perceptible deviations of $0.25°$, the simulator correctly predicts the final position of the block (Figure 3a,b). This further underscores the ability of the learned model to handle sharp discontinuities and multi-modal dynamics without producing smoothed interpolations of the training data.

## 4.2   Quantitative accuracy comparisons

In this section, we measure the performance of Graph Network Simulators on two established cube tossing datasets from the robotics community designed to capture challenges in rigid-body dynamics: a real-world cube tossing dataset from Pfrommer et al. [2], and a MuJoCo-simulated cube tossing dataset from Parmar et al. [10]. We also investigate performance on a subset of the more complex Kubric MOVi-A dataset [37].

**Experiment setup**   The datasets from [2, 10] provide the center position and the rotation of the cube at each time point, $T^{t_k}(\cdot)$. We convert these data into the positions for each node $M^{t_k}$, as our model operates over nodes instead of center/rotation. We follow the experimental setup from [2, 10], and compute the following metrics: *Positional error*: mean-squared error (MSE) over the position of the cube center as a percentage of the cube width; *Rotational error*: the angle between the ground-truth rotation and the prediction in degrees; *Penetration*: the distance of cube penetration through the floor as a percentage of the cube width; all averaged over the simulation trajectory. Following [2, 10], we train the model on a subset of trajectories from the dataset and show how the error on the test set changes as a function of the training set size.

**Real cubes**   Pfrommer et al. [2] introduce a real cube tossing dataset, providing cube positions and orientations at each time step. We compare Graph Network Simulators to the implicit ContactNets

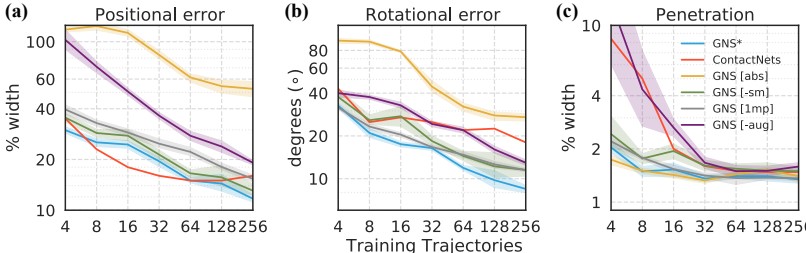

Figure 5: Impact of different architectural choices for GNS. Lack of relative encoding (yellow) causes unstable predictions; the method is however relatively robust to other changes, with ablations decreasing performance but still allowing for better predictions than ContactNets at 256 trajectories.

model (which uses rigid-contact specific priors [2]) trained on this real data. We find that the GNS outperforms ContactNets across all the presented metrics: rotational and position error, as well as penetration depth. Perhaps more surprisingly, we find that the GNS outperforms carefully hard-coded physics engines (Drake [17], PyBullet [18] and Mujoco [19]) in predicting the outcomes of held-out real cube tosses. To compare the real data to these engines, Acosta et al. [12] used system identification to determine the best fitting parameters for each engine to match the real cube tossing experiment. Their results are shown as the dashed horizontal lines in Figure 4 (penetration was not reported). The GNS trained on real cube data is able to outperform all analytic simulators in predicting position and rotation (requiring 256 trajectories for superior position prediction, and as few as 64 for rotation prediction). This suggests that the GNS, in certain circumstances, may be a better choice for real-world transfer than physics simulation engines. For example, a GNS may be preferable to circumvent error-prone system identification, or to adapt to eccentricities of the real world setup which may not be explicitly modeled by analytical simulators.

**Ablations** In order to understand the contribution of different components of the Graph Network Simulator, we conduct a series of ablations (Figure 5) on this dataset. These include encoding and predicting absolute positions on node positions instead of encoding relative displacement and predicting accelerations (GNS [abs]), removing shape matching (GNS [-sm]), limiting message-passing steps to 1 (GNS [1-mp]), and removing data augmentation (GNS [-aug]). GNS [abs] shows the largest difference: without the bias for inertial motion, the model has to make predictions on a much larger dynamic range. Breaking translational equivariance also increases overfitting, leading to unstable rollouts. Intriguingly, almost all ablations still perform better than or comparably to ContactNets on the Penetration depth metric, reinforcing the observation that graph networks can learn to resolve rigid collisions without explicit contact priors using very few training trajectories. Further ablations can be found in Appendix C.

**Robustness to noise** Finally, we performed two experiments to investigate the robustness of the GNS model to different types of noise. First, to simulate imperfect shape recovery from sensors, we add Gaussian noise with standard deviation $\sigma$ individually to the corners of the cube during training. There is a graceful degradation of the positional and rotational error as noise is increased (Figure 7a). Second, to simulate imperfect object tracking, we add noise to the cube's tracked

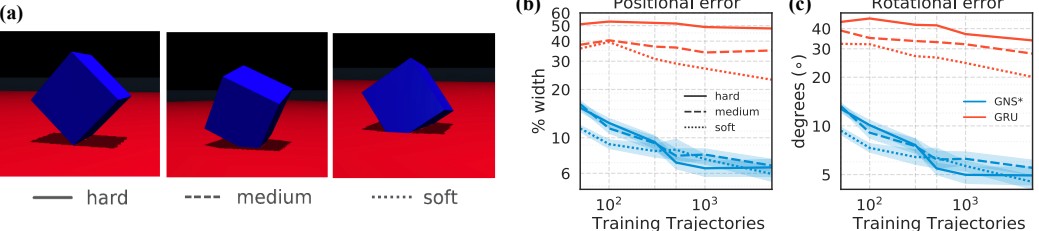

Figure 6: (a) Renders from MuJoCo with different levels of ground penetration: hard, medium and soft (Figure 3 from Parmar et al. [10]). (b), (c): performance metrics from [10] indicating positional and rotation error between the cube and the table. GNS (blue) shows significantly lower error rates than GRU [10], and better sample efficiency, for all stiffness levels.

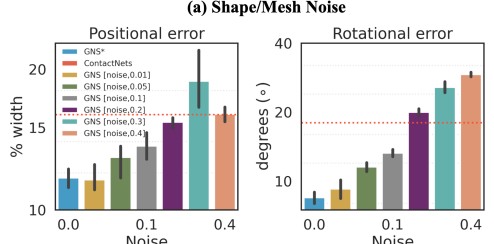

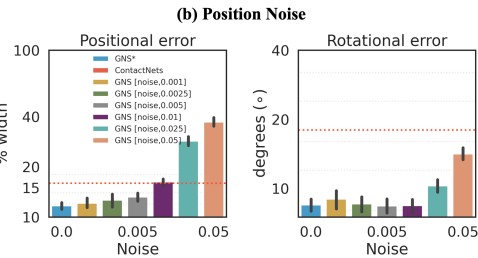

Figure 7: Testing the model's robustness to noise. (a) Shape/Mesh noise is modeled by jittering the positions of the vertices in the cube with Gaussian noise centered at 0 with standard deviation $\sigma$ (x-axis; $\sigma$ relative to cube side length of 2.0). (b) Position noise is modeled by jittering the position of the cube at each time-point in the trajectory relative to its current velocity. Noise is reported as a fraction of the cube's change in position between time-steps. Details in Appendix E.

position independently for each time-step of the trajectory during training. There is similarly a graceful degradation in the rollout error for both cube position and rotation as noise is increased (Figure 7b). In both cases, performance is better or comparable to ContactNets even for moderate amounts of noise.

**Simulated cubes with different stiffness** In contrast to other rigid-body solvers, MuJoCo uses a "soft" penalty-based contact model, with the contact stiffness determining how rigid the contacts will be. Parmar et al. [10] explicitly vary this stiffness parameter in Mujoco to investigate the ability of deep learning systems to model rigid vs. soft contacts. They show that a gated recurrent neural network (GRU) performs worse when the data is generated using stiff contacts than when it is generated using soft contacts, and note that this isolates one of the fundamental challenges for deep learning models of rigid-body dynamics.

Here we show that GNSs are not very sensitive to the rigidity with which contacts are simulated. As in Parmar et al. [10], we train the GNS on up to 10,000 trajectories (a total of 1e6 time steps) for each of the three stiffness levels provided, and compare the GNS to the GRU reported in Parmar et al. [10] in Figure 6b,c. The GNS not only outperforms the GRU, but more importantly shows little difference in its ability to model soft, medium or hard contacts, particularly as the number of training trajectories grows.

**Tossing other objects** Finally, we train a model on the Kubric MOVi-A dataset [37] which provides trajectories of complex shapes being tossed on the floor. This dataset is more diverse, consisting of three object shapes (a cube with beveled edges, cylinder, sphere) with 51–64 vertices each, two object sizes, and two sets of material properties (friction and restitution coefficients). Example rollouts are shown in Figure 8(b). For this dataset, we additionally append the friction and restitution coefficients $\sigma$ as node features, to allow the model to distinguish materials. We train a single model for all object shapes and variants, and observe that we can achieve similarly low error rates of 14.6% position error and 17.2° rotational error even in this more complicated domain (Figure 8(c)). The dataset was generated using Bullet, which uses hard collision constraints; the model can learn to replicate this behavior, showing an extremely low penetration error of 0.016% in this domain.

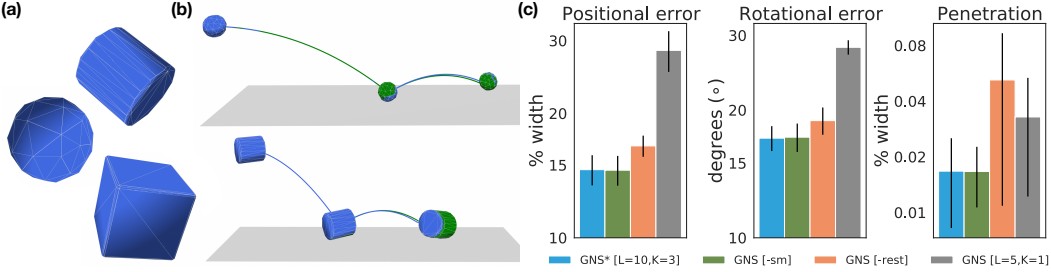

Figure 8: Tossing geometric objects with 51–64 vertices (a) from the Kubric MOVi-A dataset [37]. Even in this complicated setup with multiple shapes and friction coefficients, GNS is able to produce accurate rollouts (b, blue: prediction, green: Bullet [18] ground truth), and low error rates (c, blue).

In general, we see similar trends in ablation experiments as for the cube datasets – e.g. GNS [abs] also leads to unstable rollouts. We do however note a few differences: as seen in Figure 8(c), the model benefits from more message passing steps ($L = 10$, $K = 3$, blue) vs ($L = 5$, $K = 1$, gray), to compute collision response on these more detailed shapes. Due to the higher shape complexity, we observe some (small) benefits from including rest position features ($\mathbf{d}_{ij}^{U}$). And finally, while one might suspect that it would be harder to retain object rigidity for diverse shapes with more vertices, we observe very little shape deformation in this dataset, and model variants with and without shape matching perform equally well. This suggests that even the limited biases we built in to enforce rigidity may be lifted with more data diversity, paving the way for more general, unified simulation models involving rigid and deformable objects.

## 5 Discussion

We showed that Graph Network Simulators can precisely and accurately capture rigid-body dynamics, including discontinuities introduced by contact. A trained GNS outperforms not only other end-to-end approaches, but also other learning approaches with explicit contact assumptions [2]. Our model, trained with sufficient real-world data, is on average more accurate than real-to-sim using analytical physics engines (Drake [17], PyBullet [18] and MuJoCo [19]), matched to the same real-world experiment setup. In simulation, GNSs perform equally well on data simulated with variable levels of stiffness, and can also model other shapes, like cylinders, spheres, and beveled cubes.

We found that all aspects of the graph network simulator were necessary for best performance, especially when few trajectories were available. Remarkably, with as few as 8 real world trajectories, graph network simulators performed on par with a tuned MuJoCo model for predicting held-out real world samples (64-256 trajectories were required to outperform Bullet [18] and Drake [17]). This suggests that, at least in certain circumstances, graph network simulators may actually be a more appropriate model of real-world physical dynamics than specialized hard-coded simulators.

The effectiveness of graph network simulators for rigid contact stems from two factors: (1) operating on the explicit mesh geometry, which makes it easier to reason about contact and learn effective collision detection, and (2) a set of very general physical inductive biases, such as inertial motion and translational equivariance. Notably, these biases are not specific to rigid bodies, allowing methods with similar architectures to be applied to e.g. deformable objects. Contrary to prior claims that deep models struggle with discontinuous dynamics, Graph Network Simulators seem able to model discontinuities in rigid body motion after training on very few sample trajectories. We emphasize that there are still settings for which one might want to learn simulation models that exactly respect analytical contact equations, as in [2, 11, 13]. However, for real-world data with dynamics which are hard to model analytically, or estimate parameters for, Graph Network Simulators are an attractive option for accurately learning dynamics end-to-end.

**Limitations** In this work, we focused on demonstrating that generic Graph Network Simulators can accurately represent discontinuities arising from rigid-body collisions, but there are other aspects to rigid-body dynamics as well. It would be interesting to train GNS models on larger, more complex datasets beyond tossing single cubes, cylinders and spheres, including more complex shapes, multiple interacting rigid objects, and complex robotic manipulation or locomotion data.

Robots rarely have access to precise state information (here, the locations and poses of all objects). Instead, they have sensors that can provide haptic or visual information. The graph network simulators presented here rely on being able to extract the states of objects in the scene. While there has been some progress coupling visual estimation with graph network simulation [38, 39, 40], it remains unclear whether such combinations could support the level of precision needed to make predictions about rigid body dynamics.

Finally, despite being good forward models of dynamics, this does not guarantee that Graph Network Simulators will be useful for planning and control. Even when analytic differentiable rigid-body physics simulators are available [41, 4, 42, 43, 13], planning through contact poses a variety of challenges, such as combinatorics of local optima [44, 45, 8, 46] which can hamper finding a globally optimal solution. While GNSs may be useful as differentiable models of more general physical phenomena, they will not solve these particular challenges from planning and control directly.

**Acknowledgments**

We would like to thank Luis Piloto for reviewing an initial version of the paper and Yusuf Aytar for helpful discussions.

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
