# OpenReview forum: "Graph network simulators can learn discontinuous, rigid contact dynamics"
_robot-learning.org/CoRL/2022/Conference — CoRL 2022 Poster_

### Official Review · Reviewer_ETfg · 2022-07-25

**Originality:** Very Good
**Technical Quality:** Very Good
**Clarity Of Presentation:** Excellent
**Impact:** 3

**Recommendation:**

Weak Accept: I recommend accepting the paper, but will not argue for my recommendation if the majority of other reviewers have a different opinion.

**Summary:**

The paper investigates the applicability of general-purpose graph neural network simulators for modelling discontinuous dynamics.

Similar to existing work, the proposed method learns to predict the dynamics of a given mesh using a graph neural network that is trained with a one-step prediction loss. The resulting model is evaluated on different simulated and real-world datasets, performing on par or better than commonly used simulators, even when trained on relatively few samples. This result contradicts the previously held notion that this kind of learned simulator is unable to model discontinuous dynamics, and thus opens various avenues for further research.

**Issues:**

- As mentioned above, Section 3.1 is very similar to the MeshGraphNet paper. Clearly separating the basic MeshGraphNet architecture and the changes/adaptations made in this paper would make it easier to distinguish previous work from the contributions of this one.
- An additional paragraph explaining these adaptations would make them easier to understand. For example, it is not quite clear why the message passing block is iterated over $K$ times per update (and what $K$ is set to, as $K=1$ in previous work), or why a history of the $h=2$ previous meshes is used instead of $h=1$ in previous work.
- It should be clarified why some of the real-world tosses were removed from the dataset (as mentioned in Appendix A).
- In the appendix, it is shown that data augmentation hurts performance when training on a lot of demonstrations. An additional sentence or two on why this may be the case would be useful for understanding and validating the choice of data augmentation.
- The last paragraph of Section 5 before the `Limitations' hypothesizes why the used graph network simulators are able to model discontinuous dynamics. This hypothesis should not be stated as fact, as it is not directly proven in the paper.
- In Figure 4, it would be interesting what the penetration value is for the other solvers.
- While the experiments are generally convincing, they lack strong baselines and comparison to previous work. It would be interesting to see how the used GRU and ContactNets baselines perform on their respective other tasks. A baseline that considers velocity instead of acceleration prediction could also be useful. These baselines should also be considered for the MOVi-A dataset that showcases the behavior for the more complex shapes.

### Typos and Style

- l. 26 has two consecutive `and's
- In ll. 129-131, `position coding' is introduced. The term `positional encoding' is probably more common.
- l. 209 the `)' should come before the citation

**Quality Of The Limitations Section:**

Limitations are addressed clearly

**Reviewer Expertise:**

4: The reviewer is confident but not absolutely certain that the evaluation is correct

**Robotics Focus:**

Relevant but unlikely to deploy to hardware in near future

**Strengths And Weaknesses:**

### Strengths

- The paper shows that general-purpose graph neural network simulators for can be used to model simple discontinuous dynamics. More precisely, it shows that a slight variation of the previously proposed MeshGraphNet [1] architecture can accurately model simple rigid contact dynamics, up to a precision that matches and sometimes surpasses that of commonly used solvers. This is an interesting finding, as it contradicts the previous believe that learned general-purpose simulators are ill-suited to model discontinuous dynamics.
- The MeshGraphNet architecture is adapted to rigid body simulation by employing a shape-matching projection during the rollout phase. While the performance improvement from this additional step is minor in the measured metrics, this shape-matching is interesting as it ensures that the simulated rigid bodies can not deform over the course of a rollout.
- The experiments are thorough, and both model architecture and chosen training hyperparameters are clearly explained in the text. Experiments are repeated for $10$ seeds, and confidence intervals are reported. Different parts of the proposed method are explored through a series of ablations.
- The paper has a clear story and is well written. The figures are clean and complement the text nicely.

[1] Pfaff, T., Fortunato, M., Sanchez-Gonzalez, A., & Battaglia, P. (2020, September). Learning Mesh-Based Simulation with Graph Networks. In *International Conference on Learning Representations*

### Weaknesses

- While the results are interesting and promising, the proposed method heavily builds on previous work thus lacks novelty. Section 3.1. essentially re-introduces [1] with some minor changes here and there.
- The experiments are rather limited in scope. While they shows a clear proof of concept, it would be interesting to see how well the method works for more complex shapes.
- Very few baselines are considered in the experimental section, making it hard to directly compare the approach to existing work. This is somewhat alleviated through various ablations and comparisons to classical solvers.
- While most design decisions are ablated over, some interesting deviations from the MeshGraphNet paper remain unexplained. For example the message passing block is iterated over $K$ times, where $K=1$ in previous work.

**Summary Of Recommendation:**

The paper presents and evaluates the use of general-purposes graph neural network simulators for modelling discontinuous dynamics. It does not present much in terms of novel architectures and only considers relatively simple tasks with few baselines.

Despite these shortcomings, the paper highlights an interesting new application for graph network simulators and supports its claims with an overall thorough experimental section and various ablations.

Put together, I currently vote to accept the paper since it gives a clear new insights in a field that is not yet well understood.

---

> ### Author Response · Authors · 2022-08-26
> **Response to Reviewer ETfg Part 1**
>
> Thank you for the thoughtful review. We address your comments in-line below:
>
> > As mentioned above, Section 3.1 is very similar to the MeshGraphNet paper. Clearly separating the basic MeshGraphNet architecture and the changes/adaptations made in this paper would make it easier to distinguish previous work from the contributions of this one. An additional paragraph explaining these adaptations would make them easier to understand. For example, it is not quite clear why the message passing block is iterated over K times per update (and what K is set to, as K=1 in previous work), or why a history of the h=2 previous meshes is used instead of h=1 in previous work.
>
> We will update the methods section for the camera-ready as the reviewer suggests to make the distinctions clearer. The model always uses K=1, except for in the Kubric experiments where we found that the combination of [K=3, L=10] performed better than [K=1, L=5]. Increasing K allows the model to use a smaller number of parameters while still having long-range communication, which we thought might be helpful for both data- and computational- efficiency reasons. However, for the rebuttal we performed ablation experiments to check this (attached PDF, Section 1.4). For Kubric, which has a significant amount of training data, we found that [K=1, L=30] (which preserves how far information can propagate, but uses 3x more weights) is just as performant as [K=3, L=10]. We similarly tested sensitivity to the history length (attached PDF, Section 1.3) and found that history lengths of 2 modestly but consistently outperform history lengths of 1 for both the Real Cube and Kubric datasets. Longer history lengths may allow the model to learn a higher order integrator, while also improving robustness to noise, although this is speculation.
>
> > It should be clarified why some of the real-world tosses were removed from the dataset (as mentioned in Appendix A).
>
> From the ContactNets paper (Pfrommer et al, 2020): “After post-processing the original 750 tosses, the collected dataset contains 570 unique, high-quality tosses”. No further details on why 180 tosses were removed were provided as far as we could tell. We did not remove any of the trajectories ourselves from the provided dataset. We will clarify this in the appendix.
>
> >In the appendix, it is shown that data augmentation hurts performance when training on a lot of demonstrations. An additional sentence or two on why this may be the case would be useful for understanding and validating the choice of data augmentation.
>
> Data augmentation hurts performance specifically for the ContactNets baseline, not for the graph net model, although we do not know exactly why. We expect that this could be because of the analytic assumptions of ContactNets, where augmenting existing data may slightly violate expectations (especially when rotation augmentations are performed for transitions close to the ground), but we are not certain.
>
> > The last paragraph of Section 5 before the `Limitations' hypothesizes why the used graph network simulators are able to model discontinuous dynamics. This hypothesis should not be stated as fact, as it is not directly proven in the paper.
>
> While we do not prove this theoretically, our empirical results suggest that this is the case (and see additional experiment in Section 1.2 where we compare to an ablation which removes the mesh-based representation to further support this conclusion). We chose ablation experiments to underscore this observation. Nevertheless, we will soften the wording to make it clear that it is a hypothesis in the camera-ready.
>
> > In Figure 4, it would be interesting what the penetration value is for the other solvers.
>
> As our sysID results were taken from Acosta et al [12], we cannot evaluate the penetration value for the analytic solvers in exactly the same way. However, in Figure 5 from Acosta et al [12], they show typical penetrations for an example toss under each simulator, which appear to be between 2.5 - 10% of the cube width at maximum. We would therefore expect these results to be comparable to ContactNets and the Graph Network if evaluated over all tosses.

---

> > ### Author Response · Authors · 2022-08-26
> > **Response to Reviewer ETfg Part 2**
> >
> > > While the experiments are generally convincing, they lack strong baselines and comparison to previous work. It would be interesting to see how the used GRU and ContactNets baselines perform on their respective other tasks. A baseline that considers velocity instead of acceleration prediction could also be useful. These baselines should also be considered for the MOVi-A dataset that showcases the behavior for the more complex shapes.
> >
> > Which strong baselines would you suggest? We compare baselines only on the appropriate datasets. For example, the GRU baseline cannot be directly applied to the Kubric setting, as it cannot really learn a single model over multiple different shapes (which the GNS does).
> >
> > We have added a baseline that uses velocity rather than acceleration for both the Real Cube and MoVi-A datasets, please see the attached PDF, Section 1.1. In both cases, this baseline performs worse than using acceleration prediction.

---

> > > ### Comment · Reviewer_ETfg · 2022-08-27
> > > **Reply to the authors**
> > >
> > > Thanks to the authors for the clarification and the additional experiments. Most of my concerns have been adressed in the reply and the provided PDF.
> > >
> > > > Data augmentation hurts performance specifically for the ContactNets baseline, not for the graph net model, although we do not know exactly why. We expect that this could be because of the analytic assumptions of ContactNets [...]
> > >
> > > I greatly appreciate the honesty of the authors. I believe that data augmentation may be an important part in moving forward with GNS, and would generally be interested in finding out in which cases it would be most beneficial and why. However, I am aware that this is out of the intended scope of the paper and feel that the given explanation is sufficient for this work.
> > >
> > > > Which strong baselines would you suggest? We compare baselines only on the appropriate datasets. For example, the GRU baseline cannot be directly applied to the Kubric setting, as it cannot really learn a single model over multiple different shapes (which the GNS does).
> > > > We have added a baseline that uses velocity rather than acceleration for both the Real Cube and MoVi-A datasets, please see the attached PDF, Section 1.1. In both cases, this baseline performs worse than using acceleration prediction.
> > >
> > > I think that using e.g., a voxel-based simulator as an additional baseline would strengthen the results of the paper, as it would (ideally) showcase how the arbitrary spatial resolution of the GNS outweights the stronger inductive bias of a convolutional approach. For the scope of the rebuttal however, adding velocity baselines is fully sufficient.

---

### Official Review · Reviewer_8UbN · 2022-07-25

**Originality:** Very Good
**Technical Quality:** Excellent
**Clarity Of Presentation:** Excellent
**Impact:** 4

**Recommendation:**

Strong Accept: I recommend accepting the paper and will argue for my recommendation even if other reviewers hold a different opinion.

**Summary:**

The paper presents a learning-based approach to predicting object dynamics using graph neural networks. Object meshes are encoded using the finite difference of the mesh vertex velocity (over a short history) and distance to a boundary (e.g. floor), and edge features are relative displacements between nodes. Message passing is used to update node and edge features and the network predicts finite difference accelerations for every node. Extensive experiments are provided on simulated and real datasets of simple objects being tossed, and comparisons are made to both learned baselines and common physics engines used in robotics. The results suggest the graph network approach outperforms these baselines in terms of accuracy of predictions with relatively little training data.

**Issues:**

I would like for the authors to elaborate on the role of uncertainty in the model predictions, as I mentioned in the Strengths/Weaknesses section above. In particular that there is some limit to how accurate the predictions can be due to stochasticity from the environment, and this will be exacerbated in the real world since there is also the added uncertainty from noisy and incomplete observations.

Here are some very minor issues:
Lines 108-113: It would be good to clearly state here how to interpret the "node"/"edge" terminology with respect to a mesh, presumably that a node is a vertex of the mesh and an edge between them means they are part of the same face?
Line 265: "models involving rigid and deformable objects." (make rigid non-plural)
Line 295: Is "rigids" a term of art? It sounds awkward to me, I would go with "rigid objects".

**Quality Of The Limitations Section:**

Limitations are addressed clearly

**Reviewer Expertise:**

3: The reviewer is fairly confident that the evaluation is correct

**Robotics Focus:**

Highly relevant to robotics but no hardware experiments

**Strengths And Weaknesses:**

Strengths: The paper was clearly written and enjoyable to read. The representation of meshes in the graph network is intuitive, and Figure 2 was quite helpful in quickly understanding what was being proposed. The experiments are very thorough in that they compare to a variety of relevant baselines on a variety of datasets. The expected ablations on the proposed method are performed and provide some insight on the necessity for the different components incorporated in training the GNN. The authors identify many of the relevant shortcomings of the method including making predictions for more complex objects and multi-object scenarios.

Weaknesses: While the paper is relevant to robotics, there is minimal involvement of robots in the paper. Even the real-world dataset used from the ContactNet paper appears to just have a human tossing the cube. I find this limiting for robotics because then there is no action modality associated with any of the data, meaning it will be hard to learn predictive models for planning and control. There is also limited discussion on uncertainty and stochasticity in the dynamics, which is particularly important for tossing objects. For example, tossing the object 10 times in the exact same way will have a different outcome all 10 times, i.e. they'll vary at least a little bit due to slight variations in the execution and minute differences in the interactions with the environment. These sorts of dynamic manipulations are very sensitive to these subtle differences and can sometimes produce drastic differences in outcomes, particularly for more dynamic manipulations like dropping from higher heights, e.g. an object dropped on a table might bounce and come to rest on the table or fall off onto the floor, even when dropped from the same height in the same way. These dynamics are not fully predictable without infinite modeling precision, which is of course not possible. So it seems important to address matters of uncertainty in the predictions, and is worth discussing in more depth in the paper. This is also related to the first point about involving robots, one can control the actions to be more repeatable using a robot, which should make quantifying the uncertainty more feasible on a real-world setup, instead of having a human tossing objects for data collection. I am also curious how the approach can be applied to partial-view models, which is a very common situation in robot manipulation. The robot may only have a partial-view point cloud of the object and not have access to a ground truth mesh.

**Summary Of Recommendation:**

Making accurate predictions for object dynamics is undoubtedly relevant to robotics, and the paper does a very good job providing convincing evidence for their approach over learned baselines and common physics engines in robotics. My only reservation is that the explicit connection to robotics is tenuous. For this method to be useful for a robot, it needs to connect to the robot's action space, and none of the datasets have information about an action modality so far as I can tell. That being said, I think this paper motivates the use of a straightforward GNN architecture for simulating object dynamics, and I believe it could be a very useful tool for roboticists to build off of, which is why I'm strongly advocating for acceptance.

---

> ### Author Response · Authors · 2022-08-26
> **Response to Reviewer 8UbN**
>
> Thank you for the thoughtful comments and review. We respond to your questions below.
>
> >I would like for the authors to elaborate on the role of uncertainty in the model predictions, as I mentioned in the Strengths/Weaknesses section above. In particular that there is some limit to how accurate the predictions can be due to stochasticity from the environment, and this will be exacerbated in the real world since there is also the added uncertainty from noisy and incomplete observations.
>
> Thank you for noting this important missing aspect of the work for a robotics paper. We will add a paragraph to the discussion section specifically speaking to uncertainty and stochasticity for the camera-ready version, and have performed two further experiments to this point (see the attached PDF, Section 3).
>
> In the first experiment, we simulate “shape noise” by adding Gaussian noise with 0 mean and standard deviation sigma individually to the nodes of the cube (of side length=2.0) as a way to model imperfect perception. There is a graceful degradation of the positional and rotational error as the noise gets increased. However, even for an error of std=0.2 on the perceived mesh, both rotational and position error are comparable to ContactNets when given 256 real trajectories for training.
>
> In the second experiment, we simulate “sensor noise” by adding Gaussian noise with mean 0 and standard deviation sigma independently to each transition in the dataset. We scale the noise to the difference between the frames in order to simulate noise that depends on the speed of the moving cube. We find that there is again a graceful degradation in performance, with positional error matching ContactNets up to noise values of 0.01.
>
> We would like to acknowledge that none of these are “partial meshes” per se – the method is not well defined in that case, and it is a very interesting direction for future work. Furthermore, as the reviewer notes, while we can handle sensor noise to some extent, the loss function is still deterministic and uni-modal (being just mean squared error). This means that if the noise is sufficiently high, such that two states are *indistinguishable* but lead to different next-step states which are *distinguishable*, the model will struggle with this. Experimenting with loss functions that can support multi-modal predictions is a very interesting direction for future work.
>
> >Even the real-world dataset used from the ContactNet paper appears to just have a human tossing the cube. I find this limiting for robotics because then there is no action modality associated with any of the data, meaning it will be hard to learn predictive models for planning and control.
>
> We agree, and believe that moving to action-conditional settings is a very promising direction for future work. Prior work has shown that this is possible using graph networks for relatively simple systems [Sanchez-Gonzalez et al, ICML 2018], but as we did not have a real-world action-conditional dataset to train models for, we felt it was outside the scope of this paper. Action-conditional datasets would also require several changes to the model to indicate controlled vs. uncontrolled nodes and edges, which we believed would complicate the empirical message of the paper, but we see this work as an important building block towards this ultimate goal.
>
> > Here are some very minor issues: Lines 108-113: It would be good to clearly state here how to interpret the "node"/"edge" terminology with respect to a mesh, presumably that a node is a vertex of the mesh and an edge between them means they are part of the same face? Line 265: "models involving rigid and deformable objects." (make rigid non-plural) Line 295: Is "rigids" a term of art? It sounds awkward to me, I would go with "rigid objects".
>
> Thank you for catching these. We will update the paper accordingly for the camera-ready.

---

### Official Review · Reviewer_MGH6 · 2022-07-30

**Originality:** Fair
**Technical Quality:** Good
**Clarity Of Presentation:** Very Good
**Impact:** 3

**Recommendation:**

Weak Accept: I recommend accepting the paper, but will not argue for my recommendation if the majority of other reviewers have a different opinion.

**Summary:**

This paper proposes to use a Graph Neural Network (GNN), similar to the mesh-based graph network proposed in [3], to learn rigid-body dynamics with contact in 3D. Following [3], each mesh vertex of the rigid body is modeled as one node of GNN, and each edge of the mesh as one edge of GNN. The authors argue that using spatial-equivariant relative displacement between nodes, as in [3], and predicting acceleration rather than velocity (same as [3] in the cloth case) are important to achieve high accuracy. The method is validated both with simulated datasets [10] and a real dataset [2], and outperforms recent baselines.

**Issues:**

- L477 we “expect higher errors for later timesteps due to error accumulation”: could the authors share the final-frame errors in addition to average errors for ContactNet dataset? It’s okay if the improvement becomes less obvious compared with “those heavily tuned analytical simulators“.

- L244: the ablation study is odd to me. GNS[abs] shows that using relative displacement between nodes, *and* predicting acceleration rather than velocity are important, since it is much worse than other ablations.

   Could the authors split this important ablation into two? That is, what if we use relative displacement as input features but for output still predicts velocity/displacement, instead of acceleration? In [3], predicting acceleration and use it in second-order integration, instead of first-order seem to be an optional feature that can be easily turned on or off, which should be the same for this paper.

- Could the authors also plot the accuracy metrics on training trajectories of ContactNet (i.e. training losses), in the format of Figure 3?

- Is the Gaussian random noise (L463) identical to [16], or actually more similar to [3] which modifies upon [16]?

- Since the paper does not have much theory or insight but only demonstrates good performance, for reviewers to trust the results, could the authors promise to publish their code if the paper is accepted?

- If I understand correctly, Figure 3 (c) essentially says that if we know the initial state of a “dice”, this GNN can almost always predict correctly which face it lands on the table (otherwise rotational error would be quite large for that trajectory?), given the chaotic nature of a dice, could the authors share some insights on why you think this is not too good to be true?

- Typo in abstract: Line 5 continuous -> discontinuous?

**Quality Of The Limitations Section:**

Limitations are addressed clearly

**Reviewer Expertise:**

4: The reviewer is confident but not absolutely certain that the evaluation is correct

**Robotics Focus:**

Sufficient demonstration on hardware

**Strengths And Weaknesses:**

The strengths and weaknesses of the paper are both quite clear.

Strength:

- Simple algorithm and network structure, without heavy engineering and tuning.

- Good results on both simulated and real datasets that outperforms baseline learning methods.

Weaknesses:

- Lack of novelty: not really a weakness in my opinion, if GNN works well already out of the box, that is worth sharing with the community. But I have to say the method and implementation details are both quite similar to [3].

- Lack of insight: It’s hard for me to intuitively understand why GNN works much better than ContactNet and naive learning methods using e.g. GRUs. There is a diagnostic example (Sec. 4.1) but it does not provide hypothesis or theories why GNN is better at modeling discontinuities than *other neural nets* either.

**Summary Of Recommendation:**

“GNNs are better at modeling discontinuities than other NNs” is certainly an unconventional, and to me counterintuitive, claim. As such, since the paper do not provide much insight on why, and the testing datasets are both relatively small, I’m a bit worried about whether findings from this paper can be applied to more complex rigid-body learning problems, or larger datasets.

That being said, the results seem solid, and I would like to see these results published if the following issues can be addressed during rebuttal.

---

### Official Review · Reviewer_ehVb · 2022-07-31

**Originality:** Good
**Technical Quality:** Good
**Clarity Of Presentation:** Very Good
**Impact:** 3

**Recommendation:**

Weak Accept: I recommend accepting the paper, but will not argue for my recommendation if the majority of other reviewers have a different opinion.

**Summary:**

This paper empirically exams the question of whether graph neural newtworks can learn discoutinous, rigid contact dynamics. Specifically, the GNS [1] architecture is studied on the following discountinous contact dynamcis: tossing a cube to the ground in simulation (with different stiffness parameters), tossing a cube to the table in the real world, and tossing objects with different geometris to the ground in simulation. The experimental results show clear constrast to previous conclusions that it is difficult for GNNs to learn such rigid contact dynamics, instead, learned GNN models can perform well on such dynamcis, even better than tuned analytic physics simulators.

[1] Sanchez-Gonzalez, et al, Learning to Simulate Complex Physics with Graph Networks, ICML 2020

**Issues:**

Please see the strength/weakness section above.

**Quality Of The Limitations Section:**

Limitations are addressed clearly

**Reviewer Expertise:**

4: The reviewer is confident but not absolutely certain that the evaluation is correct

**Robotics Focus:**

Sufficient demonstration on hardware

**Strengths And Weaknesses:**

Strength:
- The question itself is interesting and the results help shed more light into the anwser -- GNN can indeed learn discountinous contact dynamcis well on the tested tossing tasks.
- The ablation studies are complete and provide good understanding of which are the important factors for GNNs to perform well.

Weakness:
- My major concern is: is the result really surprising and makes enough contribution to a CoRL paper? In the original DPI [1] paper, the learned GNN model already demonstrates correct behaviour (i.e., non penetration and correct predicted cube pose) on modeling the contact interaction between the rigid cube and the water particles (the BoxBath experiment); and in the original GNS [2] paper, the GNN model also demonstrates correct behaviour (i.e., non penetration and correct predicted particle positions) between particles colliding with environment obstacles (the WaterRamps experiment). Although these are not strictly rigid objects colliding with another rigid object, but I would not be surpised that the GNN model should be good to handle the non-peneration part of rigid-to-rigid collision based on the results from these previous papers. With this being said, these previous papers did not study if GNN can learn the sensitivity to initial conditions for rigid-to-rigid collision, which is clearly demonstrated in this paper. But with this conclusion alone, I am not entirely sure it is enough contribution for a CoRL paper.
- Also, most of the experiments considers tossing a rigid object into another. There are many more rigid contact dynamics other than this, e.g., a quadruped robot doing locomotion on the ground, a robotic hand manipulating a rigid object. Conducting some experiments in these more complicated senarios would make the result more convincing and stronger.
- Another minor question: regarding figure 4, how is the sysID done for the physics simulators? More specifically, how many real-world trajectories are used to tune the simulation parameters? I would imagine with more real-world data used, the parameters can be tuned more accurate, i.e., the performance of the simulator in figure 4 should also be a curve instead of a horizontal line. More details to clarify this would be appreciated.

[1] Li, et. al., Learning Particle Dynamics for Manipulating Rigid Bodies, Deformable Objects, and Fluids, ICLR 2019
[2] Sanchez-Gonzalez, et al, Learning to Simulate Complex Physics with Graph Networks, ICML 2020

**Summary Of Recommendation:**

Please see the strength/weakness section above.

---

### Meta-Review · Area_Chair_urob · 2022-08-11

**Recommendation:** Accept (Poster)
**Confidence:** 4

**Metareview:**

This paper provides empirical studies to show that a GNN architecture can provide better contact behavior modeling for rigid bodies than other deep learning architectures and, sometimes, can be better than physics-based methods.

All the reviewers agree that the paper poses a powerful and interesting hypothesis, and showed convincing evidence that it is true, and agree that the work is highly relevant to robotics. Many of the reviewers praise the clarity of the presentation and thoroughness of the empirical studies.  Several reviewers appreciate the simplicity of the architecture.  The reviewers also noted some concerns, but the authors addressed these concerns well in their rebuttal.  However, the authors are highly encouraged to open source their code for the final paper for the sake of reproducibility, and to encourage other researchers to build on their work.

In summary, this work is a welcome addition to CoRL, and will give valuable insights into the potential for GNN-based contact physics modeling in robotics in the future.